# Antibiotic-Induced Treatments Reveal Stress-Responsive Gene Expression in the Endangered Lichen *Lobaria* *pulmonaria*

**DOI:** 10.3390/jof8060625

**Published:** 2022-06-12

**Authors:** Tania Chavarria-Pizarro, Philipp Resl, Theresa Kuhl-Nagel, Aleksandar Janjic, Fernando Fernandez Mendoza, Silke Werth

**Affiliations:** 1Systematics, Biodiversity and Evolution of Plants, Faculty of Biology, LMU Munich, Menzingerstraße 67, 80638 Munich, Germany; philipp.resl@uni-graz.at; 2Institute of Biology, University of Graz, Universitätsplatz 2, 8010 Graz, Austria; fernando.fernandez-mendoza@uni-graz.at; 3Helmholtz Center Munich, German Research Center for Environmental Health, Institute for Network Biology (INET), Ingolstädter Landstraße 1, 85764 Neuherberg, Germany; theresa.kuhl@helmholtz-muenchen.de; 4Anthropology and Human Genomics, Faculty of Biology, LMU Munich, Großhaderner Straße 2-4, 82152 Planegg-Martinsried, Germany; janjic@biologie.uni-muenchen.de

**Keywords:** symbiotic organisms, differential expression, transcriptomic, toxic environments

## Abstract

Antibiotics are primarily found in the environment due to human activity, which has been reported to influence the structure of biotic communities and the ecological functions of soil and water ecosystems. Nonetheless, their effects in other terrestrial ecosystems have not been well studied. As a result of oxidative stress in organisms exposed to high levels of antibiotics, genotoxicity can lead to DNA damage and, potentially, cell death. In addition, in symbiotic organisms, removal of the associated microbiome by antibiotic treatment has been observed to have a big impact on the host, e.g., corals. The lung lichen *Lobaria pulmonaria* has more than 800 associated bacterial species, a microbiome which has been hypothesized to increase the lichen’s fitness. We artificially exposed samples of *L. pulmonaria* to antibiotics and a stepwise temperature increase to determine the relative effects of antibiotic treatments vs. temperature on the mycobiont and photobiont gene expression and the viability and on the community structure of the lichen-associated bacteria. We found that the mycobiont and photobiont highly reacted to different antibiotics, independently of temperature exposure. We did not find major differences in bacterial community composition or alpha diversity between antibiotic treatments and controls. For these reasons, the upregulation of stress-related genes in antibiotic-treated samples could be caused by genotoxicity in *L. pulmonaria* and its photobiont caused by exposure to antibiotics, and the observed stress responses are reactions of the symbiotic partners to reduce damage to their cells. Our study is of great interest for the community of researchers studying symbiotic organisms as it represents one of the first steps to understanding gene expression in an endangered lichen in response to exposure to toxic environments, along with dynamics in its associated bacterial communities.

## 1. Introduction

Antibiotics are instrumental in medicine and cattle farming. Several organisms can produce antibiotics as a strategy to control population sizes of harmful microbes in their environment, such as the mold fungus *Penicillium chrysogenum*, from which the first antibiotic was discovered, penicillin. Even though antibiotics are also naturally produced in ecosystems [1], the antibiotics in the environment are mainly produced by human activities, causing discharges from aquaculture and wastewater [2,3]. It has been reported that biomass production and nutrient transformation, which are important ecological functions in soil and in aquatic ecosystems, are highly disrupted by antibiotics [3,4,5]. Nonetheless, antibiotic effects on other terrestrial ecosystems have not been well studied. 

Damage associated with exposure to high levels of antibiotics has been reported in several organisms, such as muscle and liver damage in zebra finch [6], trout [7,8] and rats [9]. Previous studies have established that antibiotics could damage the genetic information of a cell, as an indirect outcome of oxidative stress or genotoxicity [10,11,12,13,14]. Furthermore, studies have also shown that antibiotics inhibit DNA damage repair in the green alga *Raphidocelis subcapitata* (Sphaeropleales) [15,16]. 

Corals previously treated with antibiotics suffer tissue loss and a substantial decrease in photosynthetic efficiency because they lack their symbiotic microbiome [17]. In the parasitic wasp *Asobara tabida*, the elimination of their symbiont bacteria *Wolbachia* using antibiotic treatment influenced their reproduction efficiency, as female wasps were completely incapable of producing viable oocytes [18]. This effect has also been found in *Coccotrypes*
*dactyliperda*, as females treated with antibiotics were incapable of laying eggs [19]. 

Lichens are traditionally described as self-sustaining symbioses of fungi and algae/cyanobacteria which develop a symbiotic phenotype known as the lichen thallus. The photosynthetic partners, the photobionts, fix carbohydrates (and some fix atmospheric nitrogen) and the fungal partners provide shelter to their photobionts between their peripheral layers, which optimizes photosynthetic functioning [20]. In the past years, it has been revealed that lichens are specifically associated with a high diversity of bacterial communities [21,22,23]. Recent studies have suggested that the lichen-associated microbiome could have the potential to increase fitness and survival rates of lichens, as they could provide aid with vital processes such as nutrient supply and defense against toxic abiotic factors [21,24,25,26,27,28,29,30]. The lung lichen, *Lobaria pulmonaria* (L.) Hoffm. (Peltigerales, Ascomycota), has more than 800 associated bacterial species and its bacterial microbiome has been hypothesized to be a third partner in the lichen symbiosis [22]. 

The tolerance of environmental stress, for example tolerance of exposure to antibiotics in the environment, allows individuals and populations to persevere in changing environments [31,32,33,34,35,36,37,38]. Cellular stress response mechanisms are highly conserved across prokaryotic and eukaryotic domains, and they include cellular antioxidation and osmoregulation, repair of damaged DNA and refolding or degradation of proteins [33,39]. The lichen-forming fungus *L. pulmonaria* responds to sudden temperature increases from 5 to 25 °C by upregulating and differentially expressing some heat shock genes [40]. In addition, Chavarria-Pizarro et al. [41] studied the effect of several factors on gene expression of *L. pulmonaria* and its green-algal primary photobiont, *Symbiochloris reticulata*. They found that both symbionts upregulated heat shock genes when there was an unexpected increase in temperature. However, whether other environmental factors such as exposure to antibiotics influence gene expression of the lichen, and if there is an interaction between responses to antibiotic treatments and to temperature increases, remain to be investigated. 

We hypothesized antibiotic treatments could represent a stress factor for *L. pulmonaria* by both their genotoxicity properties and because they could change the lichen’s associated bacterial communities, as has been previously found in other organisms with a symbiotic microbiome. For those reasons, we analyzed the mycobiont and photobiont transcriptomes of *L. pulmonaria* samples which had been exposed to antibiotic and temperature treatments to better understand their possible impact on gene expression of the lichens and their associated bacteria. Our main objectives were (1) to identify differential gene expression of lichen samples treated with different antibiotics vs. controls; (2) to analyze differential gene expression of lichens exposed to antibiotics and thermal increases to test the hypothesis that control samples exhibited less severe stress responses at high temperatures than antibiotic-treated samples; and (3) to investigate changes in microbiome abundance before and after antibiotic treatment. To our knowledge, this is the first study to assess the effects of lichen exposure to antibiotic treatments and possible consequences on symbiotic microbiome composition and to quantify gene expression differences of different symbionts. 

## 2. Methods

### 2.1. Study Species

The lung lichen *Lobaria pulmonaria* is an epiphytic species which is rare or locally extinct in most regions of Central Europe, except in some Alpine regions [42]. The mycobiont of *L. pulmonaria* associates with a green-algal primary photobiont, a species belonging to the Trebouxiaceae, i.e., *Symbiochloris reticulata* (Tschermak-Woess) Skaloud, Friedl, A.Beck and Dal Grande [43] and a minor cyanobacterial partner, *Nostoc* sp. (Nostocales) [44]. There is a high diversity of bacterial species associated with *L. pulmonaria*, with Proteobacteria being the predominant phylum and an especially high abundance of Alphaproteobacteria [21,22,26,27,45,46]. The *L. pulmonaria* and *S. reticulata* reference genomes and transcriptomes have been sequenced and the data have been released by the DOE-JGI Joint Genome Institute. 

During May 2019, we visited a locality at Rauher Weg (47.4283° N, 10.4213° E) in Hinterstein Valley in the Alps in southern Germany, where *L. pulmonaria* is locally abundant. We collected parts of individuals of *L. pulmonaria* with a large size (e.g., 100 cm^2^). The samples were randomly collected from different trees which were at least 2 m apart. Lichen lobes were placed in Petri dishes on filter paper which were kept with a lid on and acclimated in constant light (30–50 μmole m^−2^ s^−1^) at 4 °C in the laboratory for 3 weeks. Samples were placed randomly and were sprayed with deionized water frequently. 

### 2.2. Differential Visualization of Bacteria on Lichen Surfaces

Bacterial cells in lichen thalli were visualized using fluorescence in situ hybridization (FISH) following the protocol of Cardinale et al. [47] with hybridization and washing buffer composition as described in Alquéres et al. [48] and using chemicals from AppliChem. Briefly, fragments of lichen thalli were treated with an increasing ethanol series (50%, 80% and 96% for 3 min each) and rinsed twice with an ice-cold phosphate-saline buffer. Afterwards, hybridization was performed for 1.5 h at 41 °C in the dark with lichen fragments submerged in a buffer as described in Alquéres et al. [48] containing the probe EUB Atto-633 (633 nm, Metabion), specific for bacteria [49,50]. Subsequently, samples were treated with a prewarmed washing buffer (42 °C) and rinsed with ice-cold deionized water according to Cardinale et al. [47]. Additionally, samples were stained with the fluorescent nucleic acid stain SytoOrange81 (Thermo Scientific, München, DE). A total of 10 µL of the 1 nM working solution of SytoOrange81 in TE buffer was added to each fragment on the microscopic slide and incubated for 5 min. After removal of the solution, the samples were washed with 100 µL TE buffer for 1 min. The SytoOrange81 staining was repeated twice. 

Bacterial cells in the stained samples were investigated using confocal laser scanning microscopy (CLSM) on a Zeiss LSM 880 (Zeiss) equipped with an argon ion laser and a helium neon laser for excitation of Atto633 (633 nm) and SytoOrange81 (561 nm). Bacterial cells were observed with a C-Apochromat 63×/1.20 W Korr M27 water immersion objective. Zeiss software Zen Black Edition 2.3 SP1 FP1 (version 14.0.12.201, Zeiss, München, DE) was used for imaging. 

### 2.3. Antibiotic Treatments

After three weeks of acclimation under constant light (30–50 μmole m^−2^ s^−1^) at 4 °C, we split each collected lichen thallus into four (genetically identical) lobes and applied different treatments to each as follows. For ten days, we sprayed the antibiotic treatment once every two days, alternating between sterile deionized H_2_O and antibiotics (control: same volume of sterile deionized H_2_O only) in a surface-sterilized fume hood, and the samples were exposed to a daily hydration–dehydration cycle in the 4 °C cold chamber and to constant light. Each of the four lobes originating from a single thallus received a different treatment: antibiotic against (i) Gram-negatives and Gram-positives (Ampicillin); (ii) Gram-negative bacteria (Ciprofloxacin); (iii) mix of Ampicillin and Ciprofloxacin; (iv) sterile deionized H_2_O (control). The antibiotic treatments were performed in a fume hood as follows: antibiotics were sprayed onto lichen lobes residing in Petri dishes inside a plastic cover with an opening that allowed sample manipulation. After each antibiotic spray treatment, the samples were immediately put back in the cold room at 4 °C. Antibiotics were used at concentrations routinely utilized in lichen culture media (50 µg/mL), and approximately 500 µL was sprayed onto each lichen lobe each time. We repeated the antibiotic treatment three times during the ten-day period (Figure 1). On the day after the last antibiotic spray treatment, we took three samples of lichen tissue (25 mm^2^) from different parts of one lobe (sampling at 4 °C). The samples were frozen in liquid nitrogen and stored until processing at −80 °C. Three samples were taken from each lichen to account for intrathalline variability. Then, we transferred the lichen lobes into a growth chamber at 25 °C and we incubated them there for two hours. After this incubation period, an additional three samples were taken from the lichen lobes from a previously untouched area of the samples as detailed above and the material was flash-frozen and stored as before. The temperatures were chosen based on the recorded temperature spectrum at the sampling sites where *L. pulmonaria* is locally common and known stress conditions (for more details, see [41]). The experiment contained 4 biological replicates, thus yielding 96 samples (4 thalli × 4 treatments × 2 temperatures × 3 samples). 

### 2.4. RNA Extraction and DNase Digestion

RNA extraction was performed according to the manufacturer’s protocol with the RNeasy Plant Mini Kit (Qiagen, Stockach, Germany). We measured the RNA concentration and quality, after the extracted RNA had been eluted in RNase-free H_2_O, with a NanoDrop ND-1000 UV/Vis-Spectrophotometer (Thermo Scientific, Carlsbad, CA, USA) and a Qubit 2.0 fluorometer employing the RNA HS assay kit (Qiagen, Stockach, Germany). The RNA concentrations varied between 15 and 200 ng/µL. Then, we used DNase 1 digestion (Qiagen, Stockach, Germany) in our RNA samples to eliminate any remaining genomic DNA. All digested RNA samples were adjusted to a concentration of 10 ng/µL.

### 2.5. RNA Sequencing

RNA concentration and quality were checked using a high-sensitivity RNA chip on an Agilent Bioanalyzer. RNA sequencing was executed applying an adapted version of the prime-seq protocol [51], with some adjustments as described by Chavarria-Pizarro et al. [41]. We used the same protocol for cleaning and the same residual primers and preamplification as in Chavarria-Pizarro et al. [41] and we obtained the same final reaction volume of 50 µL, which was further amplified, cleaned and quantified using the protocol of Janjic et al. [51] (for more details see [41]). The cDNA libraries containing 96 barcoded samples were single-end sequenced on four lanes of a high-output flow cell on the HiSeq 1500 platform (Illumina) at the Laboratory for Functional Genome Analysis (LAFUGA) of Gene Center Munich, LMU Munich. The following sequencing setup was utilized: 16 bases for the cellular barcode and unique molecular identifier (UMI), 8 bases for the i7 barcode, and 50 bases for the cDNA read [41].

### 2.6. Data Analysis

We followed the analysis outline used by Chavarria-Pizarro et al. [41]. First, we used the zUMIs software [52] to process the raw fastq data files. The single-end sequence data were filtered and demultiplexed (for more details, see Chavarria-Pizarro et al. [41]). Then, the sequences were mapped to the *L. pulmonaria* mycobiont reference genome (https://mycocosm.jgi.doe.gov/LobpulSc1/LobpulSc1.home.html, accessed on 16 August 2021), as well as to the reference genome of the green-algal photobiont, *S. reticulata* (https://genome.jgi.doe.gov/portal/DicretExtDraftv2_4/DicretExtDraftv2_4.download.html, accessed on 16 August 2021). The assembled fungal and algal genomes contained 91.0 and 90.5% of the core eukaryotic genes, respectively.

Secondly, we used Samtools to convert quality-controlled bam files produced by zUMIs to fastq files using ‘bam2fq’ [53]. The tophat tool ‘gtf_to_fasta’ was used to generate fasta sequences from the *L. pulmonaria* reference genome GTF file [54]. Then, we used Kallisto version 0.46.0 [55] to generate count data for each transcript from *L. pulmonaria* and *S. reticulata* using the same parameters used by 41. We used the ‘tximport’ function from the tximport R-package [56] to import the abundance matrix generated in Kallisto into DESeq2 [57]. Then, we used the variance-stabilizing transformation implemented in the ‘vst’ function in DESeq2 to normalize the data variance across the mean [57,58] (more details are provided in Chavarria-Pizarro et al. [41]).

### 2.7. Differential Expression Analysis

We performed principal component analysis (PCA) in R version 3.6.3 using normalized counts (vst). The Adonis test (a randomization/Monte Carlo permutation test) was utilized to test for significant differences in gene expression variation between samples and individuals exposed to different temperatures and different antibiotic treatments. Differential expression analysis was conducted using the DESeq2 package in R. Then, we used the Wald test to compare differentially expressed transcripts between two groups using the same parameters as Chavarria-Pizarro et al. [41]. We carried out one Wald test to compare between temperatures (4 °C vs. 25 °C). Six different Wald test comparisons were made between the antibiotic treatments: (1) Control vs. Amp; (2) Control vs. Cipro; (3) Control vs. Mix; (4) Amp vs. Cipro; (5) Amp vs. Mix; (6) Cipro vs. Mix. These Wald test comparisons were made for the mycobiont and for the photobiont, respectively. Venn diagram figures were made using DiVenn 2.0 [59]. We uploaded the RNA-seq and DESeq2 analysis R scripts on data dryad (see https://datadryad.org/stash/share/6ObxvBWNeaKcQ0Fu3Wxse1GN3am0RNwRebaAGlyjIWU, accessed on 15 May 2022).

### 2.8. Weighted Gene Co-Expression Network Analysis

Weighted Gene Co-Expression Network Analysis (WGCNA) [60] was performed to identify clusters or gene modules of similarly expressed transcripts (for more details, see Chavarria-Pizarro et al. [41]). Then, we related the modules to the two explanatory variables temperature and antibiotic treatment using eigengene network methodology. We identified transcripts with high significance (*p* ≤ 0.05) for each of the two explanatory variables for the mycobiont and the green-algal photobiont separately (see https://datadryad.org/stash/share/6ObxvBWNeaKcQ0Fu3Wxse1GN3am0RNwRebaAGlyjIWU, accessed on 15 May 2022). The WGCNA code is included in the R script (see above).

### 2.9. Functional Enrichment Analysis

Gene ontologies of the *Lobaria pulmonaria* reference genome hosted at DOE-JGI, https://mycocosm.jgi.doe.gov/LobpulSc1/LobpulSc1.home.html, (accessed on 16 August 2021) were used to match differentially expressed (DE) transcripts with protein and GO-term information (see for more details Chavarria-Pizarro et al. [41]). When there was no GO term information available, we annotated the respective transcript manually by homology searches via the translated nucleotide BLAST algorithm (2021_10) (blastx) [61] against the Uniprot database (https://www.uniprot.org/blast/, accessed on 15 May 2022). We searched for homology of *L. pulmonaria* mycobiont transcript sequences to protein sequences available for several whole-genome-sequenced species of fungi. For the *S. reticulata* photobiont, we searched for homology of transcript sequences to protein sequences available for algae (for more details, see Chavarria-Pizarro et al. [41]). A locus was considered homologous if the e-value returned was smaller than 1.0 × 10^−5^. We only used this method for the differentially expressed transcripts found with DESeq2 analysis, if we did not find any GO term information available for those transcripts in the genome of *L. pulmonaria* and *S. reticulata* (Appendix A). The GO term gene information for the WGCNA is presented in the Appendix A. The total list of DE transcripts found with DESeq2 are available for the mycobiont and photobiont under the following link: https://datadryad.org/stash/share/6ObxvBWNeaKcQ0Fu3Wxse1GN3am0RNwRebaAGlyjIWU, accessed on 15 May 2022.

### 2.10. Analysis of Bacterial Communities

We also used the demultiplexed, filtered and cleaned sequences produced by zUMIs software [52] (see above) to obtain microbiome reads. Taxonomic and functional assignment of reads was carried out using DIAMOND version 6.21.5 [62] and MEGAN6 [63]. DIAMOND was used to carry out comparisons to a local copy of the NCBI nr protein database (accessed: July 2021) using the blastx algorithm. Because of the short read length (50 bp) obtained after trimming UMIs (see above), the e-value expectations were much higher than those in the quite stringent standard settings and needed to be adjusted (-e 10 -f 100). The resulting binary data files were meganized in MEGAN6 and compared. Additionally, a nucleotide-based taxonomic assignment was carried out using Kraken2 [64]. For the kmer-based identification, a standard Kraken2 database was used and then we reprocessed the taxonomic abundance with Bracken [65]. The taxonomic composition of each sample was explored via interactive plots produced by Krona [66]. Three tables were produced for every standard taxonomic level from domain to species using three columns: (1) Percentage of fragments covered by the clade rooted at each taxon, (2) number of fragments covered by the clade rooted at each taxon, and (3) number of fragments assigned directly to this taxon. The tabulated results were exported from MEGAN6 as Biom files and imported into R using the phyloseq [67] package. We used the abundance matrix to compare the taxonomic composition between samples and treatments and then we subsetted it for bacteria only. We compared alpha diversity measures between treatments using the Chao1 and ACE richness estimators and Shannon, Simpson and Inverse Simpson diversity estimators across the treatments using a Kruskal–Wallis rank sum test. The phyloseq Code is included in the R script (see above). We used the OTU abundance information to make a heat tree [68] of the taxonomic data, where the number of OTUs assigned to each taxon in the overall dataset is represented by a color [68].

## 3. Results

### 3.1. Summary Statistics and Overview of RNA-Seq

The sequence reads resulted in 4.0 to 10.5 million reads per sample, with an average of 7.5 M reads per sample in *L. pulmonaria*. Then, we obtained 14,228 transcripts after cleaning low-count transcripts (<10 counts per sample). In *S. reticulata*, we obtained 11,326 sequences after filtering low-count transcripts. The sequencing resulted in 3.5 to 9 million reads per sample, with an average of 6 million. 

### 3.2. Overall Expression Patterns in Mycobiont and Photobiont Transcripts

There was no discernable grouping of samples treated with the different types of antibiotics or the water controls in the mycobiont. The PCA of expression profiles of mycobiont transcripts revealed large variation (Figure 2A; PC1: 36% and PC2: 12%), which was confirmed using the Adonis test (r = 0.20, *p* ≥ 0.05). Correspondingly, in the photobiont, there was no clear grouping of samples by the type of antibiotic treatment visible in the PCA (Figure 2B) (Adonis test: r = 0.10, *p* ≥ 0.05), nor between high and low temperatures (Figure 2B; red/blue symbols) (Adonis test: r = 0.28, *p* ≥ 0.05). Finally, we found no detectable grouping of individual thalli treated with the different types of antibiotics or the water controls (r = 0.21, *p* ≥ 0.05) (Appendix A), nor temperature in the mycobiont (r = 0.28, *p* ≥ 0.05) (Appendix A). Consequently, in the photobiont we neither observed any grouping of individuals by the type of antibiotic treatment (r = 0.10, *p* ≥ 0.05) (Appendix A) nor by temperature (r = 0.18, *p* ≥ 0.05) (Appendix A, red/blue symbols).

### 3.3. Mycobiont Differential Gene Expression in Pairwise Comparisons of Antibiotic Treatments

A total of 235 differentially expressed transcripts were found in the mycobiont when comparing samples exposed to antibiotic treatments with controls. We found that overall, 110 transcripts were downregulated, and 125 transcripts were upregulated. A total of 49 transcripts had functional information available and could be used to derive functions. In the samples exposed to antibiotic treatments, the upregulated expressed transcripts were identified as Transcription Factors (TFs), oxidation enzymes, protein refolding proteins, DNA repair proteins, ABC multidrug transporters and efflux pumps, and genes encoding stress-activated proteins such as Hog 1 and NST1, NST2, NST3 (Appendix A, Figure 3, Electronic Appendix A).

### 3.4. Mycobiont Differential Gene Expression Associated with Temperature Changes, Resulting from Pairwise Differential Expression Comparisons

Differential expression analysis revealed 69 transcripts that were differentially expressed between temperature treatments (Appendix A). We found that overall, 37 transcripts were downregulated, and 33 transcripts were upregulated (see dataset included in https://datadryad.org/stash/share/6ObxvBWNeaKcQ0Fu3Wxse1GN3am0RNwRebaAGlyjIWU, accessed on 15 May 2022). For 24 of the differentially expressed mycobiont transcripts, protein sequence information was available (Appendix A). The transcripts downregulated at 4 °C when compared to 25 °C were identified as ANK, TPR and PNP domain-containing proteins as well as some transcripts associated with carbohydrate metabolism. Additionally, at 25 °C, transcripts were upregulated that encode HET domain-containing proteins, MAP kinase SskB, MSF transporters, oxidation enzymes, and a branched chain aminotransferase which reacts to temperature changes (Appendix A). 

### 3.5. Gene Clusters Associated with Thermal and Antibiotic Treatments in the Mycobiont

In the WGCNA, all 14,885 mycobiont transcripts were assigned into 25 modules, designated by color, with each module containing highly correlated genes (Figure 4A–C). These 25 modules were then merged using a threshold (height cut-off) of 0.3, which corresponded to a correlation of 0.70 or higher for merging (Appendix A). Merging resulted in 20 modules (Appendix A). The gene modules were related to the different experimental treatments to generate eigengene networks with assigned correlation values (Figure 4A,C). Of the 16 modules, 2 were negatively associated with antibiotic treatments, i.e., “yellow” (250) (R^2^ = −0.23, *p* < 0.05) and “brown” (270) (R^2^ = −0.43, *p* < 0.05). We found five gene modules positively associated with antibiotic treatments, i.e., “blue” (950) (R^2^ = 0.212, *p* < 0.05), “pink” (220) (R^2^ = 0.23, *p* < 0.05), “midnight-blue” (650) (R^2^ = 0.40, *p* < 0.05), “green” (200) (R^2^ = 0.4, *p* < 0.05), and “turquoise” (250) (R^2^ = 0.59, *p* < 0.05). Unfortunately, we did not find GO terms for most of the genes, just for some genes in the modules “yellow” and “turquoise”. According to the functional enrichment analysis, some biological processes that the *L. pulmonaria* mycobiont downregulated in response to antibiotic treatment in the “yellow” module were carbohydrate metabolic process (GO:0005975) and phosphotransferase activity (GO:0016868). In the “turquoise” module, the process that was upregulated in response to antibiotic treatment was the activity of some transmembrane transporters (GO:0016020, GO:0015171). 

We found four modules negatively associated with temperature, i.e., “magenta” (250) (R^2^ = −0.51, *p* < 0.05), “midnight-blue” (650) (R^2^ = −0.27, *p* < 0.05), “green” (200) (R^2^ = −0.33, *p* < 0.05), and “grey” (192) (R^2^ = −0.25, *p* < 0.05) (Figure 4A,C, Appendix A). According to the functional enrichment analysis, some biological processes that the *L. pulmonaria* mycobiont downregulated in response to temperature changes which were included in the “grey” module were the metabolic process (GO:0006725, GO:0008152, GO:0008152), the oxidation process (GO:0016491, GO:0004497, GO:0016491) and mitosis (GO:0051440, GO:0051443) (Appendix A). 

Our whole-transcriptome level results for the mycobiont were consistent with the results from the pairwise differential expression analysis provided by DESeq2, as there was a higher number of differentially expressed mycobiont genes due to antibiotic treatments than due to the temperature change. All DE transcripts that were found in the DESeq2 analysis were also found in the gene modules associated with antibiotic treatments in the WGCNA (see https://datadryad.org/stash/share/6ObxvBWNeaKcQ0Fu3Wxse1GN3am0RNwRebaAGlyjIWU, accessed on 15 May 2022). 

### 3.6. Photobiont Differential Gene Expression Associated with Antibiotic Treatment and Temperature Change, Resulting from Pairwise Comparisons

The green-algal photobiont had a similar response to the antibiotic treatments as the mycobiont. Altogether, 216 transcripts of the photobiont were differentially expressed in samples exposed to the different antibiotic treatments (Figure 5 and Appendix A). Transcripts identified to encode Transcription Factors (TFs), oxidation enzymes, protein refolding proteins, DNA repair proteins, MFS transporters and proteins in charge of the methylation process were upregulated in the photobiont in samples treated with antibiotic vs. controls (altogether, 80 transcripts were functionally identified) (Appendix A; Figure 5 and Appendix A). There were 52 DE transcripts in the photobiont when we compared the two temperatures (Appendix A). The eight transcripts we could identify encoded one DNA repair protein, one oxidation enzyme and some proteins containing specific domains (Appendix A).

### 3.7. Gene Clusters Associated with Antibiotic Treatment and Temperature Change in the Photobiont

All 11,326 DE transcripts were assigned to 20 gene modules, designated by color, which included merged modules with highly correlated eigengenes (Figure 6B). These 20 modules were then merged using a threshold (height cut-off) of 0.3, which corresponded to a correlation of 0.75 or higher for merging (Appendix A). Merging resulted in 18 modules (Figure 6A,C and Appendix A). These modules were related to the experimental treatments to generate eigengene networks with assigned correlation values (−1 to 1) (Figure 6A,C). Of the nine eigengene modules, we found that just two, namely “brown” (300 genes) (R^2^ = −0.25, *p* < 0.05) and “turquoise” (400 genes) (R^2^ = −0.25, *p* < 0.05), were negatively associated with antibiotic treatments (Figure 6A). The “turquoise” module identified GO terms that included genes related to ATP synthesis (GO:0005524, GO:0005524) and DNA replication (GO:0005663, GO:0006260) (Appendix A). For most of the transcripts (97%), no GO term could be identified.

### 3.8. Microbiome

To quantify community composition, amplicons were clustered into operational taxonomic units (OTUs). The majority of bacterial OTUs belonged to the phyla Proteobacteria (37%), Cyanobacteria (22%), Chlamydia (11%), Firmicutes (11%), and Actinobacteria (11%) (Figure 7 and Appendix A). The most abundant classes found were Alphaproteobacteria (18.8%), Chlamydia (12%), and Actinobacteria (8.6%) (Figure 7 and Appendix A). The most abundant orders were Nostocales (28%), Chlamydiales (18%), Clostridiales (10%), Rhizobiales (10%), and Rhodospillales (8%) (Figure 7 and Appendix A). Most of the bacteria belonged to the families Peptostreptococcaceae, Enterobacteriaceae, Acetobacteraceae, Rhodospirillaceae, and Rhizobiaceae (Figure 7 and Appendix A). 

The alpha diversity estimators Chao1, ACE, and InvSimpson showed some differences between the temperatures, especially between samples treated with Cipro and Mix (Figure 8). Nonetheless, the Kruskall–Wallis test did not show any statistical differences between the diversity estimates across the bacterial dataset (Table 1). 

## 4. Discussion

We examined the response of *L. pulmonaria* and its green-algal primary photobiont, *S. reticulata*, when they were treated with antibiotics and exposed to an unanticipated temperature increase. Our study has four major conclusions: (1) *L. pulmonaria* and its photobiont *S. reticulata* upregulated some stress-related genes and multidrug transporter efflux pumps when they were treated with antibiotics. (2) The mycobiont and green-algal photobiont had similar responses to the antibiotic treatments. (3) Temperature increases had a lower impact on the mycobiont and photobiont gene expression responses compared with antibiotic treatments. (4) Diversity estimates of bacterial communities did not differ, neither between antibiotic nor temperature treatments. 

### 4.1. Differential Gene Expression in Response to Microbiome Manipulations of L. pulmonaria

Organisms in a symbiotic relationship with the microbiome decrease their resistance to abiotic and biotic stress factors when they have been manipulated with antibiotics [69,70,71]. A study of the coral species *Porites astreoides* revealed that gene expression patterns during coral bleaching respond to loss of their symbiont rather than heat stress per se [72]. In addition, in the lichen *Peltigera membranacea*, along with the downregulation of gene expression of the lectin gene (*lec2*) at higher temperatures, there was a drop of cyanobacterial DNA repair gene expression [35]. Similarly, in coral larvae, a mannose-binding C-Type lectin in the coral was downregulated at increased temperatures, making the corals vulnerable to pathogens [73]. In contrast, lectins of symbiotic fungi have been hypothesized to facilitate communication between symbionts [74,75,76]. In addition, antibiotic-treated corals exposed to thermal stress were more susceptible to *Vibrio shiloi* infections and bleaching [70]. Furthermore, reef-transplanted corals that were subjected to antibiotics bleached and eventually experienced higher mortality than transplanted control corals which were antibiotics-free [71]. In addition, exposure to antibiotics had detrimental effects on survival and juvenile development of the copepod *Nitocra spinipes*, as it caused alterations in their associated microbiome [69]. 

As discussed above, we assumed that antibiotic treatments could lead to lichen genotoxicity and could reduce vitality or cause shifts in species composition of lichen-associated bacterial communities. Our results show no differentiation in the bacterial communities between antibiotic treatments and controls. Nonetheless, *L. pulmonaria* and its photobiont *S. reticulata* upregulated genes associated with stress responses such as TFs, oxidation enzymes and stress-activated protein kinases, which are usually upregulated when organisms are under stress [12,33,35,39,77,78,79]. The mycobiont *L. pulmonaria* also upregulated genes encoding Hog1 and MAPK, which are common stress response signal pathways that have been found to activate in another fungi under stress conditions [33,80,81,82]. In addition, the transcription factor C_2_H_2_ and the oxidation enzymes cytochrome P450 and GMC have been found to be upregulated when fungi are exposed to stress conditions such as heat, superoxide and osmotic pressure in previous studies [33,36,41]. 

We found that several transcripts upregulated in the mycobiont and photobiont were efflux pumps. Multidrug resistance (MDR) efflux pump proteins can provide resistance to multiple antibiotics in bacterial cells [83,84]. ATP-binding cassette (ABC) pumps and major facilitator superfamily (MFS) transporters are the two major families of efflux proteins [82,84,85,86], and both types of transporters have been found in pathogenic fungi [87,88,89]. Overexpression of efflux pump genes in the plasma membrane seems to be the major mechanism responsible for the high-level resistance that some fungi have to toxic environments, e.g., to antimicrobial drugs [87,88,89]. 

Our experimental antibiotic treatments exposed the lichen samples to oxidative stress or genotoxicity, which has been previously reported for several organisms treated with antibiotics [6,7,8,9,10,11,12,13,14]. In addition, it has been reported that algae upregulate antioxidant system genes to deal with the oxidative damage inflicted by antibiotics, which inhibits algal growth and could weaken photosynthetic activity [90]. Furthermore, exposure to antibiotics also damages cellular structures and reduces their energy reserves, e.g., the content of protein, carbohydrate and lipids [90]. Upregulation of multidrug transporter efflux pumps and of stress-related genes in *L. pulmonaria* and its green-algal photobiont could be due to cell genotoxicity caused by exposure to the antibiotics, and by upregulating these genes, the organisms are trying to reduce antibiotic levels and avoid or repair damage to their cells. 

According to our data, we cannot infer that the observed upregulation of stress-related, multidrug transporter efflux pump and oxidation enzyme genes in the *L. pulmonaria* mycobiont and photobiont is due to experimental changes in the bacterial communities. However, as we predicted, the antibiotic treatments stressed the lichen, because it has been hypothesized that by diverse functions, e.g., production of specific metabolites, bacteria increase the fitness of lichens. However, the bacterial communities of the samples treated with antibiotics had similar alpha diversity and community composition. This coincided with our microscopic pictures where we were able to find bacteria even after the antibiotic treatment (Figure 1). 

### 4.2. Differential Expression Associated with Changes in Temperature in L. pulmonaria

Flexibility of gene expression patterns could provide organisms with physiological resilience, e.g., to thermal fluctuations [91,92,93,94,95]. Temperature increases affect fungi by impacting many physiological processes, the same way it has been reported in plants, where increased temperatures can lead to suppressed immunity to pathogens as higher temperature shifts the allocation of heat shock proteins from defense to heat stress responses [94,95,96,97,98,99,100]. 

Previous studies have found that *L. pulmonaria* and its photobiont *S. reticulata* upregulated heat shock genes at an increase in temperature from 4 °C to 15 °C and from 15 °C to 25 °C, respectively [40,41]. Surprisingly, here in this study, we did not find any heat shock genes upregulated in *L. pulmonaria* and its photobiont *S. reticulata* despite the fact that that the cold-acclimated samples were exposed to a sudden temperature increase. Although heat-shock proteins (Hsps) and stress-induced proteins help repair or remove heat-induced cellular damage by preventing the aggregation of denatured proteins [100,101,102,103,104,105,106], it has been found that it is costly to sustain elevated Hsp expression under chronic stress in several organisms [72]. Elevated levels of Hsp have negative consequences on fertility, development and survival of the individuals, as energy costs increase as normal cell functions decrease during the stress response [105]. Downregulation of Hsps has been observed in coral larvae and adults which have been under long-term heat stress and were previously treated with antibiotics [72,106,107,108,109]. Therefore, *L. pulmonaria* and its photobiont could be under long-term stress when we made the RNA extractions, as they had been exposed for 10 days to antibiotic treatments, and for that reason we may not have been able to detect the exact moment when the organisms overexpressed Hsps. Future research can consider the time frame to characterize in detail the acclimation response of *L. pulmonaria* and its photobiont to antibiotic treatments [105]. 

### 4.3. Individuals Treated with Different Antibiotics and a Temperature Increase Reveal Plastic Gene Expression Responses at the Whole-Transcriptome Level in L. pulmonaria

Temperature and antibiotic treatments played a greater role in the response of the mycobiont of *L. pulmonaria* than in its photobiont. Therefore, it seems as if antibiotic and temperature treatments affected the mycobiont more than the photobiont at the whole-transcriptome level, or maybe the photobiont has a slower response to temperature increases. An interesting finding of the WGCNA of the mycobiont and photobiont is that transcripts related to metabolic processes, catalytic activity and genes related to DNA replication were downregulated and those related to transmembrane transporter activity were upregulated, which coincides with our results of the pairwise comparisons and with past studies which have found that *L. pulmonaria* shows high levels of transcriptome plasticity to deal with unexpected environmental changes [40,41]. 

Four main response pathways allow organisms to cope with stress: antioxidation, osmoregulation, repair of damaged DNA and refolding or degradation of misfolded proteins [33]. These are core stress responses conserved across all domains of life, prokaryotic and eukaryotic [39]. Our WGCNA found an upregulation of several transcripts related to DNA repair in response to the antibiotic treatments in the mycobiont and photobiont.

### 4.4. Microbiome

It has been documented that the symbiotic microbiome in *L. pulmonaria* carries out several functions such as stress responses and resistance to toxic abiotic factors [24,110,111,112]. For this reason, we assumed that antibiotic treatments should aggravate stress conditions for *L. pulmonaria*, reflected in its gene expression patterns, as the treatments would reduce vitality or cause shifts in species composition of lichen-associated bacterial communities which provide important functions to the lichen. Our results did not find differences between communities exposed to antibiotic treatments vs. controls, which may be attributed to a lack of selectivity in our treatments, as bacteria could be preadapted to deal with high levels of antibiotics produced by the lichen or by microbes in the surrounding environment. Some secondary compounds produced by lichens protect them against chemical and biological negative effects; some of them have antibacterial activity [1]. It has been demonstrated that *L. pulmonaria* possesses antibacterial activity against both Gram-positive and Gram-negative bacteria [1]. We may conclude that the microbial communities associated with *L. pulmonaria* are resilient to acute perturbations as they must adapt to deal with antibiotic metabolites produced by their host. In addition, we cannot exclude that we were not able to detect the exact moment when the communities changed or the bacterial population decreased, as the samples had been exposed for 10 days to the antibiotic treatments when we first sampled them. 

Antibiotic resistance mechanisms in bacteria include mutations that could decrease the binding of the drug to its target and increase expression of efflux pumps, which could diminish active bacterial growth, as bactericidal antibiotics require bacterial activity for the bacteria to be killed [113]. *Escherichia coli* cultures became tolerant to ampicillin by acquiring mutations that extended their lag phase, when populations were subjected to constant exposure to that antibiotic [113,114]. Krause and Werth (unpublished data) quantified bacterial activity in *L. pulmonaria* samples treated with different antibiotics. They found that treatments with antibiotics or with deionized water decreased the activity of the corresponding bacteria, which coincides with our microscope pictures (Figure 1). It is possible that the bacteria associated with *L. pulmonaria* were decreasing their active growth as a strategy to increase their tolerance to antibiotic exposure during our experiment. 

### 4.5. Conclusions

We have demonstrated here that antibiotic treatment has crucial effects on differential gene expression in the mycobiont and photobiont of the lichen *L. pulmonaria*. As our experiment did not lead to major differences in community composition or alpha diversity between antibiotic treatments and controls, upregulation of stress-related genes in antibiotic-treated samples could be caused by genotoxicity that exposure to antibiotics could have caused, and that the stress responses associated with antibiotic treatments are a response of the symbiotic partners to reduce damage to their cells. The lack of differences in bacterial communities between antibiotic treatments and controls could be due to the communities’ resilience to acute perturbations, as bacteria associated with *L. pulmonaria* must adapt to deal with the antibiotic metabolites produced by their host. In addition, it could be that our experimental timing did not allow us to detect the exact moment when the communities changed or decreased. 

Moreover, we did not find any heat shock genes upregulated with elevated temperature, which could confirm the high stress that our antibiotic treatment posed on the lichens, as organisms under chronic stress do not sustain elevated Hsp expression as it is associated with high energy cost. Furthermore, future studies can try to separate the costs and benefits of lichen gene expression using a time series of the response to antibiotic and temperature treatments. Our research is relevant for researchers studying stress responses in symbiotic organisms as it provides insights on gene expression of multiple mutualistic symbionts. It also provides a first baseline to understand gene expression in an endangered lichen in response to exposure to toxic environments, along with dynamics in its associated bacterial communities. 

## Figures and Tables

**Figure 1 jof-08-00625-f001:**
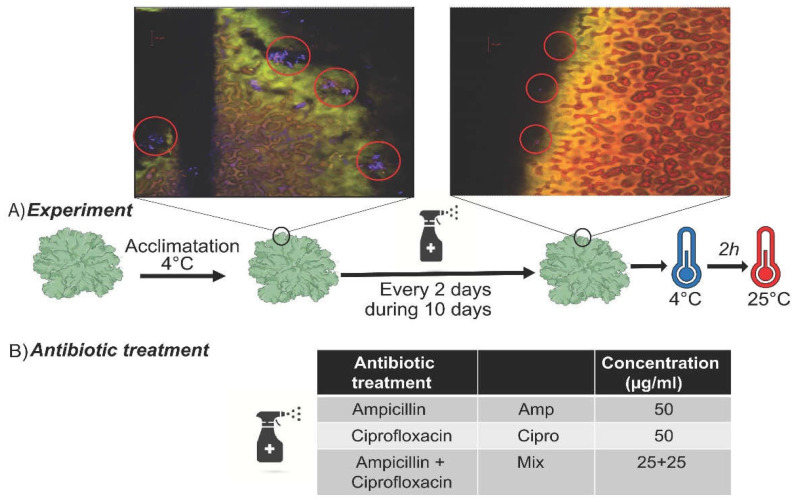
(**A**) Experimental design of antibiotic treatment: *Lobaria pulmonaria* samples were acclimated to 4 °C for three weeks. After acclimation, samples were treated with different antibiotics + deionized water (treated) or just deionized water (control), treating the lichens on every second or third day for a total of 4 treatments. Thereafter, we took samples exposed to 4 °C. After 2 h of incubation at 25 °C, a second set of samples was taken. Red circles in the figure show areas with bacteria. (**B**) Types of antibiotics used in the experiment, their abbreviations and concentrations.

**Figure 2 jof-08-00625-f002:**
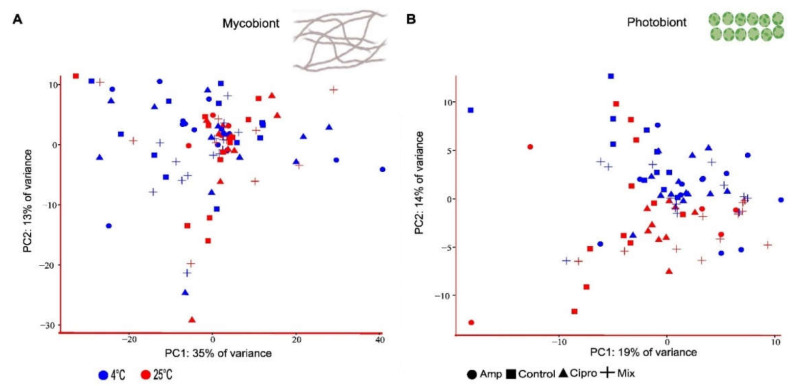
Principal component analysis of gene expression count data for *Lobaria pulmonaria* included in the antibiotic experiment (see Figure 1). Color-coding represents the temperatures that the samples were exposed to: blue (low temperature, 4 °C), and red (high temperature, 25 °C), symbols represent different treatments. Amp, Ampicillin; Mix, Ampicillin + Ciprofloxacin; Cipro, Ciprofloxacin. (**A**) Mycobiont (**B**) Photobiont (n = 96 samples).

**Figure 3 jof-08-00625-f003:**
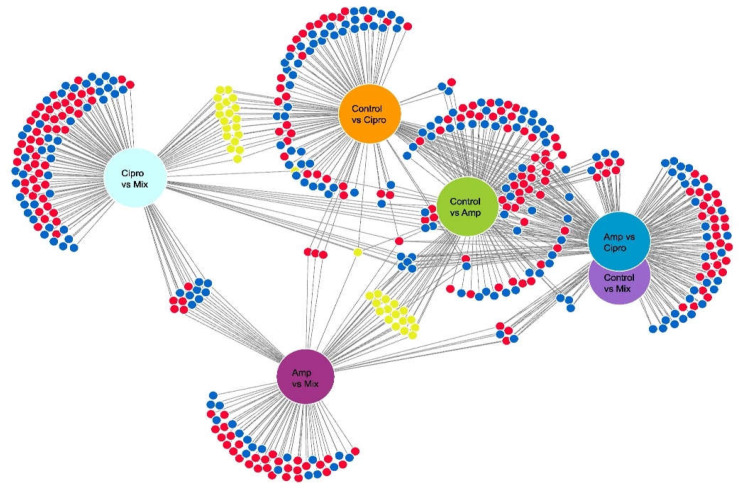
Venn diagram showing the number of differentially expressed genes in the mycobiont of *Lobaria pulmonaria*, identified by pairwise Wald test comparisons for antibiotic treatments. Downregulated: blue, upregulated: red, up/downregulated: yellow circles. Thresholds for inclusion were log_2_-fold change (>1.5=upregulated; <−1.5=downregulated) and Benjamini–Hochberg adjusted *p*-value (*p* < 0.05).

**Figure 4 jof-08-00625-f004:**
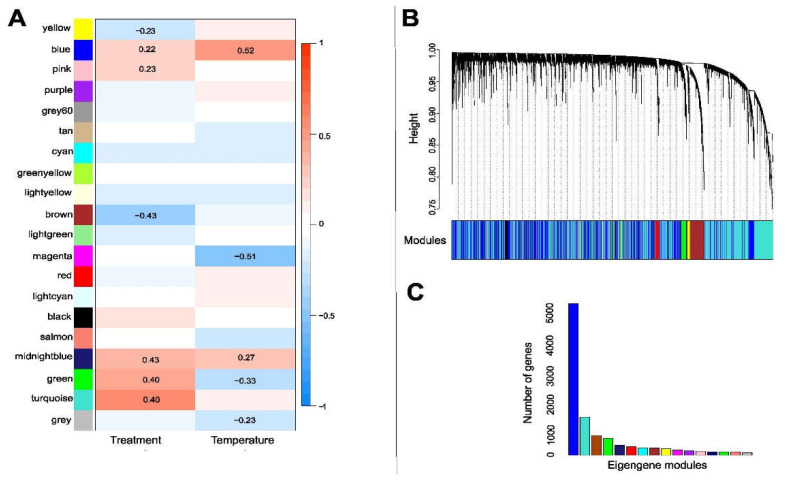
(**A**) Module traits in the mycobiont of *Lobaria pulmonaria*. The correlation coefficients for each intersection of the corresponding row and column modules are given. The red and blue shades and numbers within the boxes represent –log (P), the correlation coefficient of significant correlations according to the color legend on the right. Modules are represented by the different colors on the left side. There were seven modules associated with antibiotic treatment and five modules associated with temperature (*p* < 0.05). (**B**) Co-expression network analysis of different samples in the mycobiont of *L. pulmonaria***,** showing a hierarchical cluster tree of co-expression modules based on WGCNA. Each leaf in the tree corresponds to an individual gene. The branches correspond to modules of highly interconnected genes. The y axis represents the level of intramolecular connectivity, genes that are located at the tip of the module branches have higher interconnectedness with the rest of the genes in the module. The differently colored bars below the dendrograms indicate module membership. (**C**) Number of eigengenes per module, with colors as in (**A**,**B**).

**Figure 5 jof-08-00625-f005:**
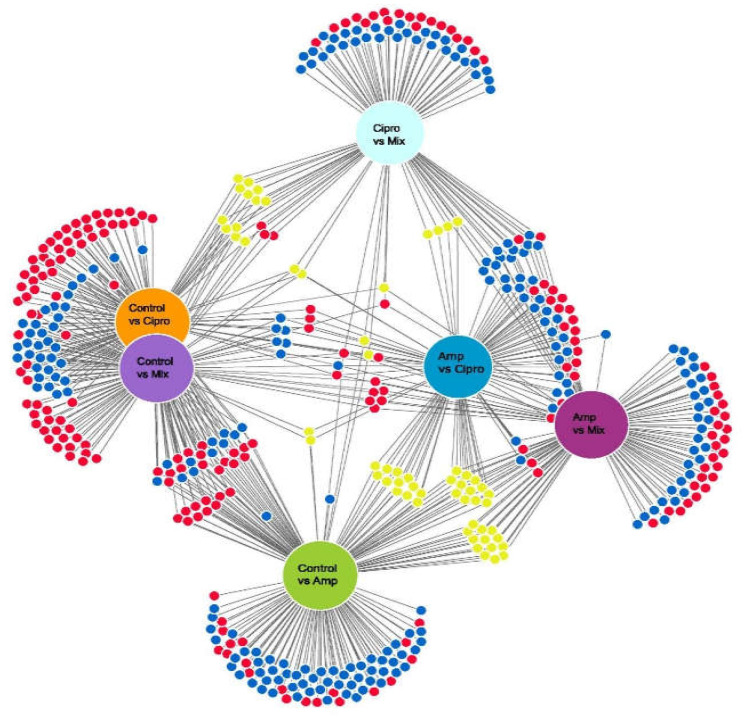
Venn diagram showing the number of genes differentially expressed genes in *Symbiochloris reticulata*, the green-algal photobiont of *Lobaria pulmonaria*, identified by pairwise Wald test comparisons for antibiotic treatments. Downregulated: blue, upregulated: red, up/downregulated: yellow circles. Thresholds for inclusion were log_2_-fold change (>1.5 = upregulated; <−1.5 = downregulated) and Benjamini–Hochberg adjusted *p*-value (*p* < 0.05).

**Figure 6 jof-08-00625-f006:**
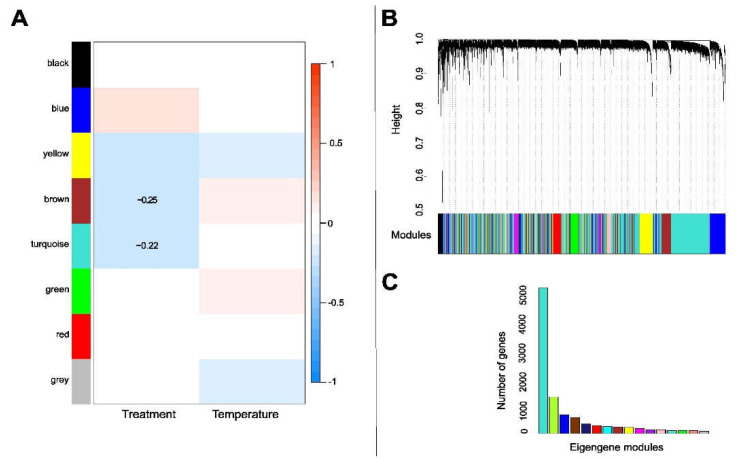
(**A**) Module traits in *Symbiochloris reticulata*, the photobiont of *Lobaria pulmonaria*. The correlation coefficients for each intersection of the corresponding row and column modules are given. The color in boxes represents –log (P), the correlation coefficient of significant correlations according to the color legend on the right. Modules are represented by the different colors on the left side. Two modules were associated with treatment (brown + turquoise) (*p* < 0.005). (**B**) Co-expression network analysis of different samples in the photobiont of *L. pulmonaria***.** Hierarchical cluster tree showing co-expression modules based on WGCNA. Each leaf in the tree corresponds to an individual gene. The branches correspond to modules of highly interconnected genes. The y axis represents the level of intramolecular connectivity, and genes that are located at the tip of the module branches have higher interconnectedness with the rest of the genes in the module. The differently colored bars below the dendrograms indicate module membership. (**C**) Number of eigengenes per module.

**Figure 7 jof-08-00625-f007:**
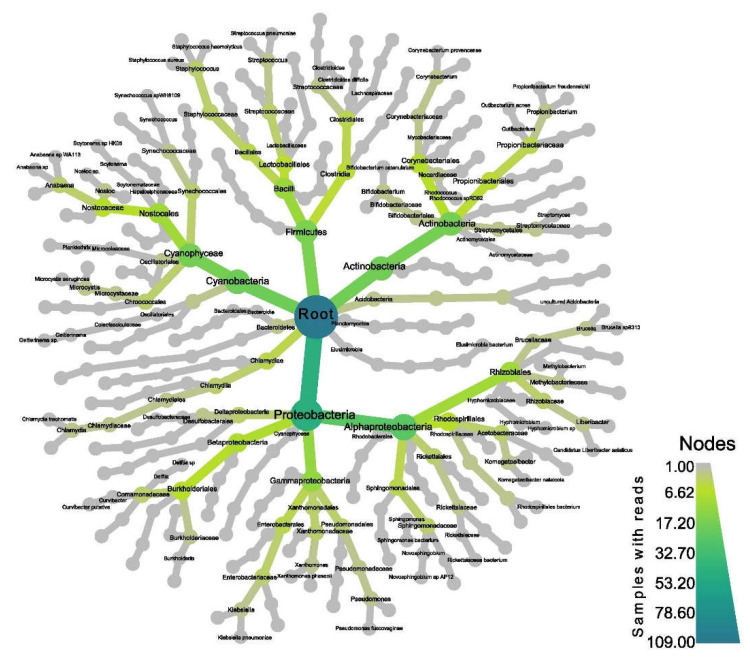
Heat tree of microbiome community structure found in *L. pulmonaria* samples. Size and color of nodes and edges show their abundance in the microbial community.

**Figure 8 jof-08-00625-f008:**
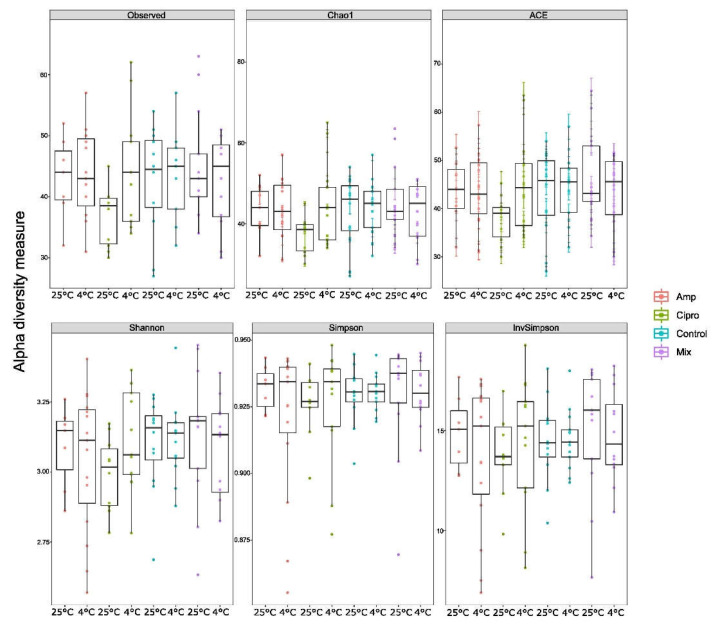
Alpha-diversity, as assessed by estimators of richness (Chao1, ACE) and diversity (Shannon, Simpson) in *Lobaria pulmonaria* samples treated with different antibiotics and temperatures. Median estimates were compared across cohorts using the Kruskal–Wallis test (*p* > 0.05). Boxes represent the interquartile range, lines indicate medians, and whiskers indicate the range of data.

**Table 1 jof-08-00625-t001:** Heteroscedasticity in diversity estimates for the bacterial dataset, as assessed with Kruskal–Wallis rank sum test.

	Observed	Chao1	ACE	Shannon	Simpson	InvSimpson
Parameter	7	7	7	7	7	7
Statistic	7.31	7.32	7.60	5.83	3.58	3.58
*p*-value	0.39	0.39	0.36	0.56	0.82	0.82

## Data Availability

The data have been deposited under the BioProject accession link (https://submit.ncbi.nlm.nih.gov/subs/bioproject/SUB11135292/, accessed on 16 August 2021). The mycobiont and photobiont total DE transcript list, the WGCNA total transcript list, the R Scripts and additional supplementary information have been deposited in data dryad (https://datadryad.org/stash/share/6ObxvBWNeaKcQ0Fu3Wxse1GN3am0RNwRebaAGlyjIWU, accessed on 15 May 2022).

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
