# Peer review of "Antibiotic-Induced Treatments Reveal Stress-Responsive Gene Expression in the Endangered Lichen Lobaria pulmonaria"

_jof, 2022, doi:10.3390/jof8060625_

Round 1

Reviewer 1 Report

The paper by Chavarria-pizarro et al. has been very improved after review. As the authors argue, experiments should include "adequate timeline in order to detect immediately changes in the bacteria communities and expression of heat shock genes", but  their  "research is relevant as we said above provides a first baseline to understand gene expression in an endangered lichen in response to exposure to toxic environments, along with dynamics in their associated bacterial communities.

I agree with them and I reccomend this paper for publication in JoF.

However, I also reccomend to simplify Figs. 4 and 6. Both Fig 4B and Fig 6B are described as a tree with branches. I cannot see any branch in those figures, only a lot of parallel lines going from a baseline to a top-line. Because the information of those figures is very shortly commented in the main text, I propose to delete both of them in the final version.

Reviewer 2 Report

Article

Antibiotic Induced Treatments Reveal Stress Responsive Gene Expression in the Endangered Lichen Lobaria pulmonaria

Tania Chavarria-Pizarro 1,*, Philipp Resl 1,2, Theresa Kuhl-Nagel 3, Aleksandar Janjic 4 , Fernando Fernandez Mendoza 2 and Silke Werth 1,*

The topic is very interesting. The paper is well written. The introduction contains clear objectives. Methods contain the most important information.

However, the following details need rewording:

line 125: „acclimated to 4 °C at 30-50 PAR in the laboratory” – should be revised (PAR is not a unit of measure),

[cf. line 150: „constant light (30-50 μmole m-2 s-1) at 4 °C,”]

The results are well presented and illustrated.

Discussion and conclusion could be somewhat improved.

Some minor mistakes are indicated directly in the text.

The manuscript should be accepted after minor changes.

The references do not follow the standard numbering and formatting of the journal. Please check authors’ instruction. Check also all links – how to cite (I did not indicate all of them in the text).

missing from the References:

Skaloud et al. 2016

not cited in the text:

Abrahamsson T. R., H. E. Jakobsson, A.F. Andersson et al. 2012. Low diversity of the gut microbiota in infants with atopic eczema. Journal of Allergy Clinical Immunology, 129, 434–440. https://doi.org/10.1016/j.jaci.2011.10.025

Alonso-Monge R., E. Roman, D.M Arana et al. 2009. Fungi sensing environmental stress. Clinical Microbiology Infection, 15, 17-19. https://doi.org/10.1111/j.1469-0691.2008.02690.x

Armaleo D., O. Müller, F. Lutzoni et al. 2019. The lichen symbiosis re-viewed through the genomes of Cladonia grayi and its algal partner Asterochloris glomerata. BMC Genomics, 20, 605. https://doi.org/10.1186/s12864-019-5629-x

Bisgaard H., N. Li, K. Bonnelykke et al. 2011. Reduced diversity of the intestinal microbiota during infancy is associated with increased risk of allergic disease at school age. Journal of Allergy Clinical Immunology, 128, e641–e645. 699

https://doi.org/10.1016/j.jaci.2011.04.060

De Vera J-P. & S. Ott. 2010. Resistance of symbiotic eukaryotes. Survival to simulated space conditions and asteroid impact cataclysms. Symbioses and Stress: Joint Ventures in Biology, Vol. (Seckbach J & Grube M, eds.), pp. 597-611. Springer, New York.

Francino M.P. 2016. Antibiotics and the human gut microbiome: Dysbioses and accumulation of resistances. Frontiers in Microbiology, 6, 1543. https://doi.org/10.3389/fmicb.2015.01543

Gauslaa Y & K.A Solhaug. 1999. High-light damage in air-dry thalli of the old forest lichen Lobaria pulmonaria - interactions of irradiance, exposure duration and high temperature. Journal of Experimental Botany, 50, 697-705. 761

https://doi.org/10.1093/jxb/50.334.697

Li J. & J. Buchner. 2013. Structure, function and regulation of the Hsp90 machinery. Biomedical Journal, 36, 106-117. https://doi.org/10.4103/2319-4170.113230

Liba C. M., F. I. Ferrara, S. Manfio et al. 2006. Nitrogen-fixing chemo-organotrophic bacteria isolated from cyanobacteria-de-prived lichens and their ability to solubilize phosphate and to release amino acids and phytohormones. Journal of Applied Microbiology, 101, 1076–1086. https://doi.org/10.1111/j.1365-2672.2006.03010.x

Lindner M., M. Maroschek, S. Netherer et al. 2010. Climate change impacts, adaptive capacity, and vulnerability of European forest ecosystems. Forest Ecology and Management, 259, 698–709. https://doi.org/10.1016/j.foreco.2009.09.023.

Pannewitz, S., B. Schroeter, C. Scheidegger, and L. Kappen. 2003. Habitat selection and light conditions: a field study with Lobaria pulmonaria. Bibliotheca Lichenologica 86:281-297.

R Core Team. 2018. R: A Language Environment for Statistical Computing (version 3.6.3). Vienna, Austria: R Foundation for Statistical Computing.

Sancho L.G., R. de la Torre, G. Horneck et al. 2007. Lichens survive in space: Results from the 2005 LICHENS experiment. Astrobiology 7, 443-454. https://doi.org/10.1089/ast.2006.0046

Scheidegger C. & S. Werth. 2009. Conservation strategies for lichens: insights from population biology. Fungal Biology Reviews, 23, 55-66. https://doi.org/10.1016/j.fbr.2009.10.003

Wang M., C. Karlsson, C. Olsson et al. 2008. Reduced diversity in the early fecal microbiota of infants with atopic eczema. Journal of Allergy Clinical Immunology, 121, 129–134. https://doi.org/10.1016/j.jaci.2007.09.011

Werth S. & C. Scheidegger. 2012. Congruent genetic structure in the lichen-forming fungus Lobaria pulmonaria and its green-algal photobiont. Molecular Plant-Microbe Interaction, 25, 2,220-230. https://doi.org/10.1094/MPMI -03-11-0081

Wickham H. 2009. ggplot2: elegant graphics for data analysis. Springer, NY

Wong J. M., K.M. Johnson, M.W. Kelly et al. 2018 Transcriptomics reveal transgenerational effects in purple sea urchin embryos: adult acclimation to upwelling conditions alters the response of their progeny to differential pCO2 levels. Molecular Ecology, 27, 1120–1137. https://doi.org/10.1111/mec.14503

Zoller S., F. Lutzoni & C. Scheidegger. 1999. Genetic variation within and among populations of the threatened lichen Lobaria pulmonaria in Switzerland and implications for its conservation. Molecular Ecology, 8, 2049-2059. https://doi.org/10.1046/j.1365-294x.1999.00820.x

Author Response

This manuscript is a resubmission of an earlier submission. The following is a list of the peer review reports and author responses from that submission.

Round 1

Reviewer 1 Report

The manuscript contains several organizational flaws that prompted me to stop reviewing it. It appears to me that the senior authors did not screen carefully the actions of the less experienced authors: figures are mislabeled or missing; figure S1 (1?) is not described in the text and treated separately from S2 (2?!). It's not clear what the differences are. The legends are redundant relative to the graphics. The data in Tables S1 and S2 are unreadable (see my comments in the attached PDF). Which figs/tables are in the main manuscript and which are Supplementary? I stopped at the end of page 15. I am supposed to review the science. I will not review manuscripts that the authors have submitted in messy and disorganized form. I will not look again at this manuscript until all authors have done their best to submit a thoroughly checked and well-organized manuscript!

I am not ready to judge the scientific value of the manuscript, but its current presentation does not dispose me favorably towards the authors.

I am sending the same comments to authors and editors, but there appears to be no attachment option for the editors.

Reviewer 2 Report

The paper "Antibiotic Induced Treatments Reveal Stress Responsive Gene 2
Expression in the Endangered Lichen Lobaria pulmonaria" by Tania Chaavarria-Pizarro el a. provides a very complete transcriptional study on the effects of some antibiotics applied to the lichen Lobaria pulmonaria. Environmental exposure to antibiotics is a global problem originated from several anthropogenic actions and studies on their effect on key-organisms must be welcome. In this aspect, the paper promise to give an interesting information on the molecular responses of an endangered lichen to a source of anthropic environmental stress. However, this article is too long and difficult to read. Sometimes, figure legends and figure descriptions in the text are quite cumbersome and difficult to follow. The study seems to be exhaustive about gene expression, but conclusions are not very clear. Why antibiotics treatments do not produce differences in the microbiome? Are those treatment really effective? The same about heat-shock treatments. Could be that the treatments were not adequate and should be chosen alternative experimental approaches? Why was not confirmed a stress-state of the lichen, after treatments? In fact, the authors write that "We did not observe any noticeable change in their photosynthetic activity after the antibiotic spraying and none of the lichens appeared to be visually damaged", measuring photosynthesis with an Imaging-PAM fluorometer. Photosynthesis fluorescence parameters are very sensitive to stress and if they are not altered by the antibiotics, very probably the treatments are not harmful for the lichen or, at least, the lichen photobionts. Finally, I think that  the whole article must be re-written in a more concise and clear form. 

Reviewer 3 Report

The manuscript entitled "Antibiotic Induced Treatments Reveal Stress Responsive Gene Expression in the Endangered Lichen Lobaria pulmonaria" examined gene expression differences in response to antibiotic treatments and temperature in L. pulmonaria. By comparing gene expression levels and bacterial community in the mycobiont and photobiont, the study identified several stress-responsive genes of the lichen, which were up-regulated under exposure to antibiotics. Although the data analyzed in this study is substantial, I have several major issues with this work. I assume that the authors tried to identify how changes in microbial community, which was caused by antibiotic treatment and/or temperature, affect gene expression of mycobiont and photobiont. However, as described in the paper, the effect of antibiotic treatment to microbial community is minute. The authors identified a handful of stress-responsive genes that were differentially expressed between the control and treatment. However, this reviewer does not understand what are the biological meanings of these upregulated genes, although the authors wrote some speculative notes in the discussion section. Moreover, this reviewer does not fundamentally understand how the antibiotics, ampicillin, alter gene expression of eukaryotic organisms, and, even if so, what is its importance in lichen symbiosis. This manuscript is poorly written, especially in its organization and displaying items (figures and tables). I couldn’t find Figure 1. Why were all the figures labelled as Figure S#. Are these supplementary figures? The authors need to pay attention for this matter before submitting the paper. The color codes in Fig. S1 disguises its meaning, and the different treatment has to be color-coded like in Fig. S2. The treatment (Amp, Con., Cip., and Mix) do not appear to be distinct from each other. Overall, the manuscript requires significant improvements as a whole to be considered for publication. Thus, this reviewer recommends not to publish this manuscript in Journal of Fungi.